# Soluble Biomarkers with Prognostic and Predictive Value in Advanced Non-Small Cell Lung Cancer Treated with Immunotherapy

**DOI:** 10.3390/cancers13174280

**Published:** 2021-08-25

**Authors:** Beatriz Honrubia-Peris, Javier Garde-Noguera, Jose García-Sánchez, Nuria Piera-Molons, Antonio Llombart-Cussac, María Leonor Fernández-Murga

**Affiliations:** Medical Oncology Department, Hospital Arnau de Vilanova, Fundación para el Fomento de la Investigación Sanitaria i Biomédica de la Comunidad Valenciana (FISABIO), 46020 Valencia, Spain; honrubia_bea@gva.es (B.H.-P.); garcia_jossanc@gva.es (J.G.-S.); piera_nur@gva.es (N.P.-M.); antonio.llombart@medsir.org (A.L.-C.)

**Keywords:** lung cancer, NSCLC, anti-PD-1/PD-L1, soluble biomarkers

## Abstract

**Simple Summary:**

Immunotherapy, most notably immune checkpoint inhibitors (ICIs), has revolutionized the treatment of advanced non-small cell lung cancer (NSCLC). Although some patients respond well to ICIs, many patients do not benefit from ICIs, leading to disease progression and/or immune-related adverse events. Biological markers can help to improve patient selection. However, currently available markers such as PD-1 and its ligand (PD-L1) have important limitations. For this reason, new biomarkers obtained by non-invasive methods are urgently needed. In the present review, we describe recent advances in the development of novel soluble biological markers (e.g., circulating immune cells, TMB, circulating tumor cells, circulating tumor DNA, soluble factor PD-L1, tumor necrosis factor, etc.) for patients with NSCLC treated with immunotherapy.

**Abstract:**

Numerous targeted therapies have been evaluated for the treatment of non-small cell lung cancer (NSCLC). To date, however, only a few agents have shown promising results. Recent advances in cancer immunotherapy, most notably immune checkpoint inhibitors (ICI), have transformed the treatment scenario for these patients. Although some patients respond well to ICIs, many patients do not benefit from ICIs, leading to disease progression and/or immune-related adverse events. New biomarkers capable of reliably predicting response to ICIs are urgently needed to improve patient selection. Currently available biomarkers—including programmed death protein 1 (PD-1) and its ligand (PD-L1), and tumor mutational burden (TMB)—have major limitations. At present, no well-validated, reliable biomarkers are available. Ideally, these biomarkers would be obtained through less invasive methods such as plasma determination or liquid biopsy. In the present review, we describe recent advances in the development of novel soluble biomarkers (e.g., circulating immune cells, TMB, circulating tumor cells, circulating tumor DNA, soluble factor PD-L1, tumor necrosis factor, etc.) for patients with NSCLC treated with ICIs. We also describe the potential use of these biomarkers as prognostic indicators of treatment response and toxicity.

## 1. Introduction

In recent years, the emergence of immune checkpoint inhibitors (ICIs) has increased the life expectancy of patients with advanced lung cancer. However, their introduction into clinical practice has raised many questions with regard to the optimal treatment and delivery sequence of these antibodies [1] ICIs are humanized monoclonal antibodies that primarily target programmed cell death protein 1 (PD-1), programmed death-ligand 1 (PD-L1), and cytotoxic T lymphocyte-associated protein 4 (CTLA-4) [1]. PD-L1 is a membrane protein expressed by cancer cells, which binds to PD-1 expressed by lymphocytes (T cells, B cells and natural killer cells [NK]). The PD-1/PD-L1 axis is a major immune checkpoint control that regulates inhibitory interactions between immune and tumor cells. PD-L1/PD-1 binding inhibits immune cell activity against malignant cells, and PD-L1 expression protects tumor cells from immune attack [2,3]. Anti-PD-1 and anti-PD-L1 antibodies block this interaction, thus restoring cytotoxic immune response. However, a substantial percentage of patients will not benefit from ICIs and one of the major challenges for immunotherapy treatment remains patient selection. Although several predictive biomarkers are available, their accuracy is less than optimal.

The best known—and most widely used—predictive biomarker of response to anti-PD-1/PD-L1 treatment is PD-L1 expression, which is assessed by immunohistochemistry (IHC). Indeed, this is the only criterion used to select patients for anti-PD-1/PD-L1 treatment in clinical trials. However, tumor response can occur in patients with low or negative PD-L1 levels on IHC, and a notable proportion of patients with high PD-L1 expression do not benefit from the treatment. In addition, PD-L1 expression is influenced by spatial tumor heterogeneity [4,5,6] and temporal variation, particularly after chemotherapy [7]. Due to these limitations, there is a clear need to identify more reliable predictors and response biomarkers to improve patient selection in the clinic and in clinical trials.

In order to characterize the tumor microenvironment prior to treatment initiation and to establish the prognosis and predictive fact, sufficient quantities of tissue samples are needed for biomarker analysis. However, this is often challenging due to difficulties in reaching the tumor, small tumor size, and the time required to analyze the sample, all of which can delay treatment initiation. In this regard, these challenges could be overcome through the use of serum biomarkers in peripheral blood. At present, however, no serum biomarkers have yet been validated and approved for use in cancer patients by regulatory bodies in Europe or the United States, despite important recent advances in our technical, genomic, proteomic and metabolic understanding of cancer. Clearly, the availability of peripheral blood biomarkers in routine clinical practice would be invaluable.

In this context, there is a growing interest in identifying serum biomarkers to predict immunotherapy outcomes, which would thus overcome the limitations of tissue-based biopsy [8,9,10]. Soluble biomarkers (serum or plasma) have numerous theoretical and practical advantages over tissue biopsies, including easier access, less invasiveness, the ability to perform sequential analysis during follow-up, and greater representativeness of the tumor microenvironment. Numerous serum biomarkers are currently under investigation. Among the most promising of these are interleukins (IL), interferon gamma (IFNγ), tumor necrosis factor (TNF) [11], and soluble PD-L1 (sPD-L1) [12]. Serum levels of neutrophils, lymphocytes, and platelets have also been associated with immunotherapy efficacy and patient prognosis [8,13,14]. Other potential serum biomarkers under investigation include circulating tumor cells (CTCs) [15,16], tumor mutational burden (TMB) [17], circulating tumor DNA (ctDNA) [8], soluble Granzyme B [18], microRNA (miRNA) [19], blood microbiome [20], and exosomes [21]. All of these soluble factors can be detected and measured in plasma, which makes them good candidates as predictive biomarkers of immunotherapy (Figure 1 and Table 1).

In the present study, we review recent advances in liquid biopsy to identify potential prognostic and predictive biomarkers of response in patients with advanced NSCLC treated with immunotherapy (Table 1).

## 2. Circulating Immune Cells

Microenvironmental inflammation plays a fundamental role in the carcinogenic process and in the progression of malignant lesions. Inflammation promotes cell proliferation and survival, activates angiogenesis, and reduces response to antitumor agents. Neutrophils and other immune cells such as myeloid-derived suppressor cells and macrophages also secrete factors that promote tumor growth, including TGF-beta, VEGF, IL-6, IL-8 (proinflammatory cytokines) [11], and matrix metalloproteinases involved in angiogenesis and metastasis [22] (Table 1 and Figure 1).

Given that these immune cells are the effectors of ICI treatment, it could be advantageous to precisely calculate the numbers of these cells and to perform a detailed analysis of immune system cells present in peripheral blood. Several authors have identified neutrophilia as an inflammatory response capable of inhibiting antitumor immune response due to inactivation of T cells [13,14], which suppresses cytotoxic activity. Platelet activation is stimulated by proinflammatory cytokines and participates in the recruitment of neutrophils [14]. Several studies have found an association between the neutrophil-to-lymphocyte ratio (NLR) and overall survival (OS), progression-free survival (PFS), and overall response rate (ORR) in patients with advanced NSCLC treated with immunotherapy [13,14,22,23]. The findings of those studies suggest that all of these outcome variables (PFS, OS and ORR) are significantly worse in patients who present higher NLR values and platelet counts at baseline (prior to immunotherapy) and in the 4-to-6-week period following treatment initiation. However, these results must be interpreted cautiously given the limitations of those studies, most of which were small, retrospective studies. In addition, the cut-off points used for the NLR and the platelet rate were highly heterogenous, thus making it difficult to compare the results. For example, some studies used an NLR of 3 [22,23] whereas others used an NLR of 4 or 5 [13,23].

Lymphocytes play a central role in the antitumor response induced by immunotherapy. Therefore, determination of baseline lymphocyte status is useful to assess immune-competence and to quantify the number of specific pre-existing tumor clones [8]. High lymphocyte and eosinophil counts have been associated with a greater benefit in OS in patients with advanced melanoma treated with immunotherapy [8]. It is believed that T cells in peripheral blood also reflect the tumor microenvironment and presence of tumor infiltrating lymphocytes (TIL) [8]. Previous studies have found that neoantigen heterogeneity in tumor tissue intensifies the reactivation of T CD8+ lymphocytes in the tumor and increases the therapeutic action at the immunological control point [24]. Other studies have found that T cell receptors (TCR) that allow for the recognition of a variety of epitopes [25] are extremely similar in PD-L1+ and CD8+ T cells and TILs, indicating that PD-L1 expression in peripheral T cells could be an indicator of the immune status of the tumor tissue [24]. Dronca et al. [26], analyzed peripheral blood levels of Bcl-2-like protein 11 (*BIM*) in patients with metastatic melanoma. They observed that patients with clinical benefit after four cycles of treatment with immunotherapy presented a higher frequency of Bim+/PD-1+ CD8 T lymphocytes at the beginning of the study compared to patients with disease progression (mean of 60% vs. 49%, *p* = 0.04). In 9/9 responding patients, PD-1+ CD8 T cell BIM levels decreased after the first three months of treatment and increased or did not change in the 5/5 non-responders (*p* = 0.003). Authors conclude that Bim levels in tumor-reactive PD-1+ CD8 T cells may select patients likely to benefit from anti-PD-1 therapy. A recent study found that the TCR diversification in peripheral CD8+ T cells was associated with a better prognosis in patients treated with anti-PD-1 therapies [27]. Another study involving patients with NSCLC treated with anti-PD-1 antibodies, found higher levels of Ki67+, PD-1+ and CD8+ T cells in peripheral blood, especially among responders [28], and this increased the population of T cells that expressed co-stimulatory molecules, such as CD27, CD28 and inducible T cell co-stimulator (ICOS), with CD28 being the main target of PD-1/PD-L1 signaling [8] (Table 1 and Figure 1).

In a very recent study, Anagnostou et al. [29] performed comprehensive genomic and transcriptomic tumor analyses in conjunction with T cell repertoire analyses for 64 advanced melanoma patients treated with ICIs as part of the CheckMate-038 clinical trial (NCT01621490). The authors showed that the tumor mutation burden is associated with improved treatment response; however, the mutation frequency in expressed genes is superior in predicting outcome. Increased T cell density in baseline tumors and dynamic changes in regression or expansion of the T cell repertoire during treatment differentiate responders from non-responders subjects. In addition, transcriptomics studies showed an increased abundance of B cell subsets in tumors from responders and patterns of molecular response associated to expressed mutation elimination or retention that reflect clinical outcome. Furthermore, they showed that CD8+ T cell exhaustion, identified here through TOX expression, could be reversible and lead to a therapeutic effect for ICIs.

In addition, Wu et al. [30] showed clear evidence of clonotypic expansion of effector-like T cells not only within the tumor but also in the surrounding healthy tissue. Subjects with gene signatures of such clonotypic expansion respond best to anti-PD-L1 treatment. Remarkably, expanded clonotypes found in the tumor and in the surrounding healthy tissue can also be detected in peripheral blood, which suggests a suitable approach to patient identification. These results and external datasets suggest that intra-tumoral T cells, mainly in responsive patients, are replenished with fresh, non-exhausted replacement cells from sites outside the tumor, suggesting continued activity of the cancer immunity cycle in these individuals, the acceleration of which may be associated with clinical response.

Interestingly, soft tissue sarcomas represent a heterogeneous group of cancers, with more than 50 histological subtypes. A gene expression study (*n* = 608) of sarcomas has allowed a classification of five subtypes based on immunity from the composition of the tumor microenvironment. Subtype E (high immunity) is characterized by the presence of tertiary lymphoid structures (TLS) containing T cells and follicular dendritic cells and particularly rich in B cells. Group E demonstrated improved survival and a high response rate to PD-1 blockade with pembrolizumab in a phase II clinical trial. This work demonstrates the clinical potential of B-cell-rich ELTs for decision-making and clinical treatments with ICs, which could have further potential applications in other cancers [31].

Additionally, CD8+ T lymphocytes are the main antitumor effector cells. Most cancer immunotherapeutic strategy seeks to increase cytotoxic T lymphocytes (CTL) specific to tumoral cells. The tissue-resident memory T (TRM) cells, a recently identified subpopulation of memory CD8+ T cells, persists in peripheral tissues and does not recirculate. TRM cells subpopulation is considered an independent memory T cell lineage with particular transcription factor expression. TRM cells also express high levels of granzyme B, IFNγ and TNFα (see below), a reinforcement of their cytotoxic characteristics. In spontaneous tumor models and engrafted tumors, natural TRM cells or cancer-vaccine-induced TRM directly control tumor growth. In addition, TRM cells predominantly express immune checkpoint receptors such as Tim-3, PD-1 and CTLA-4. Blockade of PD-1 with anti-PD-1 antibodies on TRM cells obtained from human lung cancer promotes cytolytic activity toward autologous tumor cells. Therefore, TRM cells appear to represent relevant factors in tumor immune surveillance. TMR cells induction by immunotherapy strategy may be crucial for the success of these treatments. TRM cells have the following characteristics: (1) contact with tumor cells, (2) predominant expression of immune checkpoint receptors, and (3) recognition of tumor cells, which suggests that they may be involved in the success of ICIs treatment in various cancers [32]. In addition, the baseline presence of tertiary lymphoid structures and their components is correlated with the capacity of cancers to undergo intra-tumoral immune cell reactivation.

Finally, platelets play a fundamental role in systemic and local responses against cancer. They sequester tumor molecules, including RNA and protein transcripts, altering their RNA profiles. After their interaction with the tumor microenvironment, they are called tumor-educated platelets. They transport material from the tumor microenvironment to sites close to the tumor, creating favorable environments for the development of metastases. They contain a rich repertoire of RNA varieties, providing biomolecules for diagnosis and prognostic, predictive or follow-up biomarkers [33].

## 3. Tumor Mutational Burden

TMB is defined as the number of mutations per megabase (Mb) of DNA [34]. Rizvi et al. observed an association between high TMB and better immunotherapy outcomes in patients with NSCLC [35], potentially due to neoantigens induced by acquired mutations, which increase tumor immunogenicity and thus response to immunotherapy [36]. To date, no association has been observed between TMB and high PD-L1 expression, but this association has been observed in tumors presenting epidermal growth factor receptor (*EGFR*) mutations, which could explain the lack of efficacy of immunotherapy in this patient subgroup. By contrast, Kirsten rat sarcoma viral oncogene homolog (*KRAS*) and proto-oncogene *BRAF* driver mutations are associated with high TMB, with a greater response to immunotherapy [34]. In the CheckMate-227 trial, which compared immunotherapy (nivolumab plus ipilimumab versus nivolumab monotherapy) to first-line chemotherapy in advanced NSCLC, found that OS was better in patients with high TMB (>10 mutations/Mb) [37]. However, an even greater response was observed in patients with high TMB and PD-L1 expression (>50%) compared to patients who presented only a single factor, indicating that TMB may be a good independent biomarker when considered together with PD-L1 expression [36,37]. In a recent meta-analysis, Galvano et al. evaluated patients with advanced NSCLC treated with immunotherapy (eight patient cohorts in five different phase III clinical trials) in which TMB levels (high vs. low in either tissue or blood) was evaluated as a potential biomarker of efficacy. High TMB was associated with better OS in patients treated with immunotherapy versus chemotherapy [38].

Despite the positive results of that meta-analysis, TMB is still not ready for use in routine clinical practice due to its many limitations. For example, the assessment of TMB requires long sequencing panels, which in turn requires large amounts of tumor tissue [36]; by contrast, less tissue is required to analyze the expression of PD-L1 [36]. The major obstacle to implementing TMB as a biomarker in routine clinical practice is the need to standardize the available tests in order to achieve a more homogeneous approach to measuring TMB levels to ensure comparability among tests. In addition, despite the widespread use of terms such as “low TMB” and “high TMB”, the threshold to define high or low levels has not been clearly established (Table 1 and Figure 1).

Several studies have confirmed that TMB is a potential biomarker for immunotherapy in melanoma [39], lung cancer [40] and urothelial cancer [41]. However, in many patients with advanced cancer, it is simply not possible to obtain sufficient tumor tissue for molecular analysis [17], which explains the interest in identifying less invasive methods to guide immunotherapy, such as the use of circulating tumor DNA to detect TMB, known as blood TMB (bTMB). Gandara et al. [42] found that bTMB was associated with improved PFS outcomes in patients with NSCLC treated with immunotherapy versus chemotherapy; however, more evidence is needed before this approach can be used in clinical practice. Wang et al. [17] examined the correlation between bTMB and TMB in tumor tissue (tTMB) in order to assess the value of bTMB for selecting patients with advanced NSCLC that could benefit from anti-PD-1 and anti-PD-L1 therapies. Those authors found that patients with bTMB levels ≥ 6 obtained significantly better PFS and ORR. That study also showed higher levels of bTMB among responders (Table 1 and Figure 1).

In short, measuring TMB in peripheral blood through ctDNA appears to be a promising strategy to obtain an effective, independent biomarker to estimate response to immunotherapy. However, many questions remain unanswered, including the appropriate panel size and variants to include.

## 4. Circulating Tumor Cells

CTCs are cells that have escaped from a primary neoplasm and transit in the bloodstream. Their capture and analysis offers new opportunities to better understand the tumor metastasis process, potentially facilitating the development of novel cancer treatments [43]. CTCs can provide complementary data, potentially obviating the need for tumor biopsy [44], from the primary tumor and/or metastatic lesions, thus providing a global representation of the genetic diversity of the tumor [15]. Unlike ctDNA, CTCs can be analyzed at the single cell level, thereby providing real-time data on tumor heterogeneity [15]. Currently, CTCs are being studied in parallel to ctDNA and miRNA as a potential predictive biomarker in different tumors [15]. The isolation of CTCs is challenging due to their rarity (1 cell per 106 or 107 leukocytes in human blood), fragility, and heterogeneity. These characteristics make detection highly difficult. Although multiple technologies have been developed to isolate these cells, the optimal method remains unclear [15]. Punnoose et al. [45] detected *EGFR* mutations in CTCs and ctDNA, finding that the mutations present in those cells were highly concordant with the mutational state of the tumor tissue. Guibert et al. [16] evaluated whether CTCs could represent a substrate to analyze PD-L1 expression, finding that CTCs expressing PD-L1+ had no correlation with DFS or OS in patients with advanced NSCLC. However, patients with positive PD-L1 expression in CTCs ≥ 1% at baseline were more likely to be non-responders. Importantly, PD-L1 expression was evaluable in 93% of CTC samples, but only 72% of tissue samples due to the limitations of conventional biopsy [16] (Table 1 and Figure 1).

Nicolazzo and colleagues [46] monitored CTCs in patients with advanced NSCLC undergoing immunotherapy (nivolumab) in order to determine whether there was an association between PD-L1 expression in CTCs and response to immunotherapy. Although the small sample size (*n* = 10) impeded an accurate analysis, they found that the presence of PD-L1+ CTCs was associated with progression in a period of 6 months in 50% of the patients, suggesting that PD-L1-positivity in CTCs could be an indicator of resistance to immunotherapy. Nevertheless, those findings must be interpreted cautiously due to the small sample size (Table 1 and Figure 1).

In short, the presence of CTCs is associated with a worse prognosis and a higher risk of early relapse in different types of cancer. Therefore, the study of CTCs could improve our understanding of the molecular biology of tumors and also provide useful information on tumor spread. This strategy may help to identify potentially relevant molecular targets in order to improve therapeutic management. CTCs could be used to guide sequential therapy with tyrosine kinase inhibitors (TKI) in patients with NSCLC with acquired resistance mutations. Although high levels of CTCs pre-treatment seem to be associated with worse prognosis, current data indicate that PD-L1 expression status in CTCs has no prognostic impact [16,46] (Table 1 and Figure 1).

## 5. Circulating Tumor DNA

The contribution of ctDNA to the development of next-generation sequencing (NGS) has played an important role in the diagnosis and follow-up of various solid tumors. The clinical application of ctDNA has progressed significantly in melanoma and NSCLC, as the mutational status of oncogenes such as *BRAF* and *EGFR* are important factors in the therapeutic decision [47]. Analysis of ctDNA could predict response to immunotherapy given that treatment efficacy has been associated with TMB [47]. Lipson et al. [48] evaluated the correlation between ctDNA levels and clinical symptoms in melanoma patients treated with immunotherapy. In that study, in which five of the ten patients presented mutations, an increase in ctDNA TMB was associated with tumor progression. In addition, ctDNA monitoring has the potential to be used for the non-invasive detection of these mutations in patients treated with anti-PD-1 antibodies [12] (Table 1 and Figure 1).

## 6. Soluble Factor of PD-L1 (sPD-L1)

Protein PD-L1 is expressed in transmembrane and soluble forms. Elevated plasma levels of sPD-L1 have been observed in patients with various types of tumors, including lung cancer, lymphoma, hepatocellular carcinoma, and gastric cancer [49].

Based on the findings of the KEYNOTE 024 trial, PD-L1 expression in tumor cells (measured by IHC) was established as a predictive biomarker for immunotherapy in patients with NSCLC. In that trial, patients with PD-L1 expression levels > 50% in tumor cells showed a greater response to pembrolizumab than conventional treatment [50]. By contrast, in the CheckMate-026 trial, first-line nivolumab did not prolong PFS compared to chemotherapy in patients with advanced NSCLC who had PD-L1 expression over 5% [51]. These results suggest that PD-L1 alone is insufficient to assess the efficacy of anti-PD-1/PD-L1 treatments [8]. Moreover, as a biomarker, PD-L1 expression has several important limitations, most notably the lack of consensus regarding the appropriate cut-off point to define the degree of valid expression (1% being the most common threshold level). In addition, tumor heterogeneity and the difficulty in obtaining a sample with conventional biopsy, limit the interpretation of the PD-L1 tumor expression (Table 1 and Figure 1).

Several studies have evaluated the functional role of sPD-L1 in cancer, as this marker could indicate the magnitude of the intrinsic response of T cells in cancer tissues. High levels of sPD-L1 have been associated with worse prognosis; a study in 96 patients with advanced NSCLC found that high sPD-L1 levels (>7.32 ng/mL) were significantly associated with shorter OS (hazard ratio [HR], 1.99: *p* = 0.041) [12]. By contrast, another study reported that sPD-L1 levels were not associated with tumor histology, the presence of driver-mutations, or clinical stage [8].

The available evidence suggests that high levels of sPD-L1 seem to be associated with worse prognosis in patients with advanced NSCLC. However, the prognostic value of sPD-L1 expression remains uncertain. More prospective studies are needed to reliably and reproducibly correlate PD-L1 expression in tumor tissue and plasma.

## 7. Tumor Necrosis Factor, Interferon Gamma, and Interleukins

Boutsikou et al. described the important role that IFNγ, TNF, and IL play in tumor development [11]. Those authors evaluated IFNγ, TNF and IL levels in peripheral blood at baseline and after three months of anti-PD-1 therapy, finding a significant association between high levels of these cytokines and response rates and OS, but not in PFS. Cytokine levels increased upon initiation of anti-PD-1 treatment but decreased three months later. Furthermore, those authors found that PD-L1 expression was significantly associated with differences in IFNα levels but not with the other cytokines. However, the potential role of these cytokines as prognostic or predictive biomarkers of response to immunotherapy is not well understood due to a lack of data supporting these preliminary results (Table 1 and Figure 1).

Certain interleukins—including IL-4, IL-1β and IL-6—are involved in key mechanisms of carcinogenesis and assessment of these interleukins could be a promising strategy in antitumor immune therapy [11]. Most ILs are secreted by CD4+ T lymphocytes, but other cells, including monocytes, macrophages and endothelium, also produce ILs. IL-12 has been shown to interrupt the cell cycle, induce apoptosis in human hepatocellular carcinoma cells, and change the tumor microenvironment from pro-oncogenic to antitumorigenic by recruiting immune cells, which explains why cytokine concentration levels seem to be associated with PFS [11]. Tran et al. [52] found that high IL-8 concentrations were associated with worse PFS in patients with kidney cancer treated with TKIs. In a large-scale retrospective analysis (*n* = 1344), Schalper et al. [53] demonstrated that elevated baseline serum IL-8 levels are associated with poor outcome in patients with advanced cancers (melanoma, NSCLC, renal cells carcinoma) with nivolumab and/or ipilimumab, everolimus or docetaxel therapy. This study showed the significance of evaluate serum IL-8 levels in identifying unfavorable tumor immunology and as an independent biological marker in subjects receiving ICIs.

## 8. Other Soluble Biomarkers

### 8.1. Granzyme B

Granzyme B is a serine protease secreted by cytotoxic CD8+ T cells and NK cells, with an important role in immune response and induced apoptosis. Granzymes are transported to the target cells by cytotoxic granules and are responsible for caspase-dependent apoptosis. High levels of soluble granzyme B have been linked with inflammatory, cardiovascular and atherosclerotic diseases [54,55,56]. Preclinical studies have found that granzyme B activity can be used as a prognostic factor for response to ICIs [18]. A recent study [57] in 633 patients with cutaneous melanoma showed that granzyme overexpression suppresses cell proliferation and migration but not apoptosis in cutaneous melanoma cell cultures. The authors of that study emphasized the important prognostic value of granzymes in these patients, suggesting that granzymes could also predict anti-PD-1 immunotherapy response, thus potentially allowing for better selection of candidates. A current study of patients (*n* = 347) with stage IV NSCLC treated with nivolumab demonstrated that patients with low granzyme B levels had worse PFS and OS compared to subjects with elevated serum concentrations [58]. Costantini et al. [59] also found that patients who presented a positive response to nivolumab had significantly higher soluble granzyme B levels at the start of nivolumab treatment compared to non-responders. Those authors found a clinical benefit and a trend toward better ORR, OS, and PFS in patients with stable or decreasing concentrations of soluble granzyme B between treatment initiation and the first evaluation. Elevated levels of circulating granzyme B may indicate the presence of an efficient and activated cytotoxic immune cell response, mainly CD8+ cytotoxic cells, which are associated with better response to ICIs [59].

### 8.2. miRNA

Micro RNA are non-coding RNA fragments (15–25 pb) with biological activity. Its role in solid tumors has been well investigated, with evidence of involvement in tumor suppression and resistance to chemotherapy [60]. It has been suggested that miRNA is involved in immune response regulation and associated with PD-L1 and PD-1 expression, and thus of potential interest for diagnostic and therapeutic purposes. Costantini et al. [59] described the connection between downregulation of circulating miRNA-320b and miRNA-375 expression and immunotherapy response. Studies have shown that miRNA-320b is downregulated in several different tumor types; in NSCLC, this has been associated with carcinogenesis and poor prognosis [61]. Other studies have shown that miRNA-375 is downregulated and associated with poor prognosis in NSCLC [19,62]. Furthermore, miRNA-375 is closely associated with the cell signaling Wnt/β-catenin [63] and Hippo pathways [64], both of which have been implicated in immunotherapy resistance [65]. As a result, both miRNA-320b and miRNA-375 appear to be involved in anti-cancer immune response to ICIs.

Several other miRNAs (miRNA-200, miRNA-197, and miRNA-34) have been associated with PD-L1 expression in NSCLC. Fujita et al. suggested that chemoresistance in NSCLC patients with elevated PD-L1 expression may be regulated by miRNA-197 [66]. Gibbons and colleagues found that low expression of miRNA-200 was associated with high PD-L1 expression and suppression of CD8+ cells [67]. In a small study (*n* = 20), Halvorsen et al. [68] identified seven miRNAs (miR-215-5p; miR-411-3p; miR-493-5p; miR-494-3p; miR-495-3p; miR-548j-5p; and miR-93-3p) associated with longer OS in patients with NSCLC treated with nivolumab.

The evaluation of miRNA expression can provide useful data that can be applied to help diagnose NSCLC and estimate the potential value of anti-PD-1/anti-PD-L1 therapy. MicroRNA expression could even be a possible therapeutic target. To date, however, only a few studies have evaluated the correlation between miRNA and response to ICIs. Moreover, due to the limitations of those studies (e.g., small size and heterogenous analytical techniques), it is not possible to reach any meaningful conclusions regarding the potential role of miRNAs as predictors of response and survival in patients treated with immunotherapy (Table 1).

### 8.3. Gut Microbiome

Recent studies in animal models have underscored the function of the microbiota in mediating response to chemotherapeutic drugs and immunotherapy (anti-PD-L1 or anti-CTLA-4) [69]. The intestinal microbiota regulate these treatments through critical, well-structured mechanisms involving bacterial translocation, innate immunoregulation, metabolism, enzymatic activity, and decreased variety and ecological diversity (processes known as TIMER). Ouaknine et al. [20] analyzed the influence of the blood microbiome, circulating citrulline (a marker of intestinal barrier function), and early administration of antibiotics on the efficacy of nivolumab in NSCLC. That study incorporated 72 subjects with advanced NSCLC with anti-PD-L1 treatment in a second-line (or subsequent) setting. Patients who received early antibiotics had a worse OS and lower citrulline concentrations than patients not treated with antibiotics. Elevated basal citrulline rates were associated with longer PFS and OS. Those authors also evaluated blood microbiota composition according to tumor response and clinical benefit with anti-PD-L1, finding that *Gemmatimonadaceae* DNA in blood was associated with low response, tumor progression, and worse OS in patients with early antibiotic use. Paradoxically, in patients who did not receive early antibiotics, the microbiome profile was high in *Solibacteres*, which were associated with an improved treatment response compared to patients who received antibiotics. In this clinical setting, plasma citrulline and blood microbiome appear to be promising predictive biomarkers in NSCLC.

**Table 1 cancers-13-04280-t001:** Potential soluble biomarkers for cancer immunotherapy.

Biomarker	Clinical Relevance	Authors
**Circulating immune cells**	▪ High Treg count associated with longer OS ▪ Relative number of CD4+ and CD8+ cells correlate with response and longer OS. ▪ Higher PD-L1 expression in peripheral T cells associated with non-response, shorter PFS and OS. ▪ NK cells associated with response to PD-1 antibodies. ▪ NLR ≥ 3–4 associated with shorter OS and NRL ≥ 2.2 with non-response. ▪ Disadvantages: Time-consuming, labor-intensive analysis and high cost.	Mitsuhashi et al. [8] Jiang et al. [22]
**Tumoral mutational burden (TMB)**	▪ TMB defined as the number of mutations per megabase of DNA. ▪ ICI treatment: high TMB is associated with high clinical benefit. ▪ Good independent biomarker. ▪ Disadvantages: - Requires long panels for sequencing and large amounts of tumor tissue. - “Low” vs. “high” TMB not clearly defined.	Greillier et al. [34] Hellmann et al. [37] Heeke et al. [36]
**Circulating tumor cells (CTCs)**	▪ Cells that have detached from a primary tumor and circulate in the bloodstream. ▪ Provide complementary information, thus avoiding the need for tumor biopsy. ▪ Presence of CTCs is associated with a worse prognosis and a higher risk of early relapse. ▪ Disadvantages: rarity, fragility and heterogeneity.	Hong et al. [43] Muinelo-Romay et al. [44] Pawlikowska et al. [15]
**Circulating tumor DNA (ctDNA)**	▪ ctDNA are short DNA fragments derived from dying cells. ▪ Seem to indicate the response to immunotherapy because the efficacy of the treatment depends on TMB. ▪ Disadvantages: requires costly next-generation sequencing (NGS).	Cabel L, et al. [47]
**Soluble PD-L1 (sPD-L1)**	▪ sPD-L1 is expressed in transmembrane and soluble forms. ▪ High level of sPD-L1 have associated with worse prognosis. ▪ Disadvantages: - No consensus regarding cut-off point to define valid expression (1% is the most common threshold). - Correlation between sPD-L1 expression and the prognostic value remains uncertain.	Okuma et al. [12] Mitsuhashi et al. [8]
**Circulating proteins and cytokines**	▪ Soluble immune-related biomarkers. ▪ Levels of IFNγ, TNF and IL are associated with ICI benefit. ▪ Interleukins are involved in key mechanisms of tumorigenesis. ▪ High concentrations of IL-8 associated with worse PFS in kidney cancer patients treated with tyrosine kinase inhibitors. ▪ Disadvantages: Although the analysis is protocolized, more studies are needed.	Boutsikou et al. [11]
**Granzyme B**	▪ This is a serine protease secreted by NK cells and cytotoxic CD8+ cells. ▪ Essential role in immune response induced apoptosis. ▪ Elevated circulating levels manifest an efficient and activated cytotoxic immune cell response. ▪ Disadvantages: analysis is expensive (ELISA).	Larimer et al. [18] Costantini et al. [59]
**microRNA (miRNA)**	▪ Non-coding RNA fragments (15–25 pb) with biological activity. ▪ PD-L1 expression is strictly associated with miRNA function in lung cancer. ▪ Implicated in tumor suppression and chemoresistance. ▪ Disadvantages: differences in expression levels between healthy individuals and patients are usually tiny. Large sample sizes are necessary to obtain the basal line of potential marker and decide whether it could clearly segregate the health and disease status. Only those who have high sensitivity and specificity in people with different characteristics have the potential for clinical application. ▪ The blood sampling methods should be carefully considered for some specific miRNA biomarkers.	Naidu et al. [60] Chen et al. [62]
**Microbiota and microbiome**	▪ The microbiome comprises all genetic material within a given microbiota. ▪ The gut microbiome influences the efficacy of PD-1 treatment in epithelial tumors. ▪ Blood microbiome composition were determined according to tumor response and clinical benefit with anti-PD-L1. ▪ Disadvantages: few studies, larger sample sizes needed for statistical validity, high cost.	Alexander et al. [69] Huang et al. [70]
**Exosomes**	▪ Class of extracellular vesicles made and released by most cells; contains mRNA, miRNA and proteins. ▪ miRNAs in exosomes are more stable than serum miRNAs due to the double-membrane structure. ▪ Exosomes can bind to PD-1 on immune cells and thus actively suppress CD8+ lymphocytes. ▪ Disadvantages: - Methods for tumor-derived exosome isolation and separation from human plasma-derived exosomes are not yet established and their role as biomarkers remains unconfirmed. - Low purity, specialized equipment required, exosome aggregation.	Chen G et al. [21] Poggio et al. [71] Del Re M et al. [72]

**Abbreviations:** OS: overall survival; PFS: progression-free survival; NRL: neutrophil to lymphocyte ratio; ICIs: immune checkpoint inhibitors.

In a recent study using mouse models [70], microbial analysis showed that *Parabacteroides distasonis* and *Bacteroides vulgatus* were more prevalent in responders to anti-PD-1 blockade than non-responders. That study also showed that transplantation of fecal material from mice that responded to PD-1 inhibitors to non-responders sensitized response in these mice, thus converting them from non-responders to responders [70]. Takada et al. [73] studied the clinical effect of probiotics in 294 patients with advanced or recurrent NSCLC treated with anti-PD-1 immunotherapy (nivolumab or pembrolizumab monotherapy). PFS—but not OS—was significantly higher in subjects taking probiotics. Univariate analyses revealed that probiotic use was associated with significantly better disease control and overall response. These results suggest the potential value of gut microbiota as a new biomarker for predicting response to anti-PD-1 immunotherapy. Nevertheless, large, prospective studies with more ICI treatment strategies are needed to confirm those findings (Table 1).

### 8.4. Exosomes

Exosomes are membrane-bound extracellular vesicles (EV) produced in the endosomal compartment of most eukaryotic cells. EVs can transport functional proteins, mRNA, and miRNA to adjacent cells, allowing them to perform their function as the mediator of cell-to-cell interaction (Table 1 and Figure 1). Cancer cells actively release exosomes into the bloodstream, which are distributed throughout the body and become part of the complex communication system stimulated by the tumor stroma. Exosomes are present in all body fluids and have a potential role in communication between cells and non-invasive cancer biomarkers. Exosomal-PD-L1 is detectable in human plasma and could be an important biomarker. It can bind to PD-1 on immune cells and thus actively suppress CD8+ lymphocytes. Exosomes can also suppress NK cell activity, interfere with monocyte differentiation, and induce differentiation of myeloid precursor cells into myeloid suppressor cells [74]. Therefore, a higher concentration of exosomal PD-L1 before initiation of anti-PD-1 therapy has been associated with worse prognosis in melanoma patients [21,71].

Xiao-Peng et al. found that the type of miRNAs from the exosomes of NSCLC patients are different from those in healthy subjects. Those authors reported that the presence of three miRNAs of the hsa-miR-320 family could be potential indicators of the efficacy of immunotherapy in these patients. Furthermore, hsa-miR-125b-5p could be a target for anti-PD-1 treatment in plasma exosomes of NSCLC patients [75]. Del Re et al. investigated the relationship between PD-L1 mRNA in plasma-derived exosomes and response to nivolumab and pembrolizumab in patients with melanoma and NSCLC. They showed that the exosomal expression of PD-L1 changes during anti-PD-1 treatment. Furthermore, they observed that the concentration of PD-L1 from plasma exosomes decreased significantly in subjects who respond to treatment and, conversely, increased in patients with progressive disease; no relevant changes were observed in individuals with stable disease [72].

Plasma-derived exosomal miRNAs and mRNA may be potential biomarkers for anti-PD-1 treatment in advanced NSCLC. Recently, Guyon et al. [76] reported that exposure to anti-PD-1 in T cells induces enrichment of exosomal miRNA-4315, which promotes an apoptosis-resistance mechanism to standard chemotherapy in cancer cells receiving this miRNA. The authors, also establishing the molecular mechanism, found that apoptosis-resistance is associated with downregulation of a proapoptotic protein (Bim) mediated by miRNA-4315. In a very recent preclinical study, exosomes were purified from two melanoma cell lines to study the immune inhibitory effect of exosomes in in vitro models in patient-donated NY-ESO-1-specific CD8+ T cells. Exosomes from both cell lines inhibited the immune response of antigen-specific T cells in a similar manner, as demonstrated by decreased expression of the cytokines IFN-γ and TNF-α. Furthermore, the authors observed that this inhibitory effect could be partly reversed by the presence of anti-PD-L1 and anti-IL-10 antibodies [77]. The authors concluded that the inhibitory capacity of exosomes should be considered in the process of developing therapies that rely on the potency of personalized antigen-specific effector T cells. The application of plasma-derived exosomes in anti-PD-1 immunotherapy has several clinical advantages. First, the soluble exosomes detected in patients are mainly from cancer cells, which can more accurately and dynamically represent the status and function of cells and tumor tissue. Second, miRNAs in exosomes are more stable than miRNAs in blood due to the presence of the protective double membrane. Finally, for the detection of exosomal miRNAs, only one blood sample is needed, avoiding the disadvantages related to the heterogeneity of tumor tissue samples, thereby allowing for adequate monitoring of the disease course during treatment (Table 1).

## 9. Conclusions

The growing body of evidence points to the enormous potential of immunotherapy, which may one day become a cornerstone treatment for multiple types of cancer. This is why it is so crucial to develop reliable biomarkers (Figure 1). At present, research in this area is still in the early stages. Relatively few clinical studies have been performed, and many are retrospective or have other important limitations. Nevertheless, the future of immuno-oncology is highly promising, and it is expected that numerous novel biomarkers will be discovered in the coming years. Studies conducted to date show that biomarkers can and will play an important role in predicting treatment response and toxicity. Nonetheless, more targeted, prospective studies are needed to validate these biomarkers in clinical practice.

## Figures and Tables

**Figure 1 cancers-13-04280-f001:**
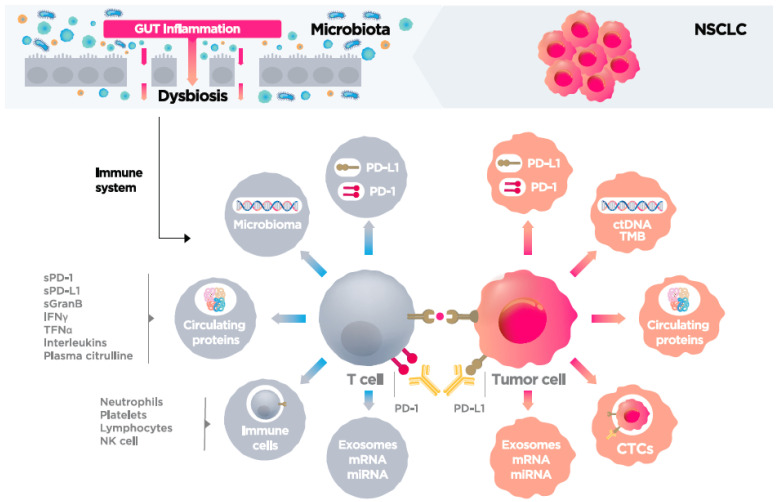
Soluble biomarkers of immune system response to immune therapy.

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
