# Peer review of "Soluble Biomarkers with Prognostic and Predictive Value in Advanced Non-Small Cell Lung Cancer Treated with Immunotherapy"

_cancers, 2021, doi:10.3390/cancers13174280_

Round 1

Reviewer 1 Report

This review 

is very much well written and it especially focuses on the limitations (the most important issue) of the actual diagnostic tools to predict responsiveness to ICI treatment.

Very small issues can be addressed and strengthen the impact of the manuscript:

  1. Paragraph 2. This sentence should be better detailed as the great relevance of the subject. “Other studies have found that T-cell receptors (TCR) that allow for the recognition of a variety of epitopes [26] are extremely similar in PDL1+ and CD8+ T cells and TILs, indicating that PDL1 expression in peripheral T cells could be an indicator of the immune status of the tumor tissue [25].”
  2. Circulating proteins, other than sPD-L1 and cytokines, have not been completely described.
  3. Include a section were a correlation between the actual approved tumor biomarker (i.e. CEA) and ICI responsiveness is made.
  4. Little is discussed about tumor-educated platelets

Author Response

Reviewer #1: s very much well written and it especially focuses on the limitations (the most important issue) of the actual diagnostic tools to predict responsiveness to ICI treatment.

We thank the reviewer#1 for the time and effort put on revising our manuscript and for the positive feedback made on its format and content.

Very small issues can be addressed and strengthen the impact of the manuscript:

1-Paragraph 2. This sentence should be better detailed as the great relevance of the subject. “Other studies have found that T-cell receptors (TCR) that allow for the recognition of a variety of epitopes [26] are extremely similar in PDL1+ and CD8+ T cells and TILs, indicating that PDL1 expression in peripheral T cells could be an indicator of the immune status of the tumor tissue [25].

Following the reviewer’s suggestion, in section “2. Circulating immune cells” we have added:

“Dronca et al [26], analysed peripehral blood levels of Bcl-2-like protein 11 (BIM) in patients with metastatic melanoma. They observed that patients with clinical benefit after four cycles of treatment with immunotherapy presented a higher frequency of Bim + / PD-1 + CD8 T lymphocytes at the beginning of the study compared to patients with disease progresion (mean of 60% vs. 49%, p = 0.04). In 9/9 responding patients, PD-1 + CD8 T-cell Bim levels decreased after the first three months of treatment and increased or did not change in the 5/5 non-responders (p = 0.003). Authors conclude that Bim levels in tumor-reactive PD-1+ CD8 T cells may select patients likely to benefit from anti-PD-1 therapy”.

2-Circulating proteins, other than sPD-L1 and cytokines, have not been completely described.

We have completed this section as suggested by reviewer #2:

““In a large-scale retrospective analysis (n=1344), Schalper et al. [53], demonstrated that elevated baseline serum IL-8 levels are associated with poor outcome in patients with advanced cancers (melanoma, NSCLC, renal cells carcinoma) with nivolumab and/or ipilimumab, everolimus or docetaxel therapy. This study showed the significance of evaluate serum IL-8 levels in identifying unfavorable tumor immunology and as an independent biological marker in subjects receiving ICIs.”

3-Include a section were a correlation between the actual approved tumor biomarker (i.e. CEA) and ICI responsiveness is made.

We welcome the reviewer's suggestion. However, classic tumor markers such as carcinoembryonic antigen (CEA) have little role in the management of patients with lung cancer and the literature on their use for diagnosis or follow-up of evolution is very scarce. Furthermore, their role as predictive biomarkers of response to treatment or prognosis in patients treated with immunotherapy. For this reason they have not been mentioned in this review.

4-Little is discussed about tumor-educated platelets

Following the reviewer’s suggestion, in section “2. Circulating immune cells” we have added:

“Platelets play a fundamental role in systemic and local responses against cancer. They sequester tumor molecules, including RNA and protein transcripts, altering their RNA profiles. After its interaction with tumor microenvironment, they are called tumor-educated platelets. They transport material from the tumor microenvironment to sites close to the tumor, creating favorable environments for the development of metastases. They contain a rich repertoire of RNA varieties, providing biomolecules for diagnosis and prognostic, predictive or follow-up biomarkers [33]”.

Reviewer 2 Report

This is a well-written and documented comprehensive review with very appropriate  references.

The authors provide a critical analysis of each of these biomarkers that is relevant.

Some suggestions to improve the impact of this review

- In the biomarkers of the tumor microenvironment, PD-L1 and TMB are well developed. TLS (Petitprez F Nature 2020) and resident memory CD8+T cells)(Karaki S et al J Immunother 2021) (Mami-Chouaib F et al J Immunother Cancer 2018)(Voabil P Nat Med 2021) could be mentioned

- At the blood level, there is a lot of recent work on Tcell -repertoire as a predictive biomarker of clinical response to immunotherapy. The authors cite the 2017 article by Akyüz. Other more recent articles could be reported (Anagnostou V CelL Rep Med 2021, Wu TD Nature 2020).

- Similarly, the authors could cite the work of I Melero on the value of IL-8 as a biomarker of response to immunotherapy (Schalper KA et al Nat Med 2020)

- Line 125-129: the sentence "Other 125 studies have found that T-cell receptors (TCR) that allow for the recognition of a variety 126 of epitopes [26] are extremely similar in PDL1+ and CD8+ T cells and TILs, indicating that 127 PDL1 expression in peripheral T cells could be an indicator of the immune status of the 128 tumor tissue [25]" could be clarified.

- It could be discussed that an important point in these biomarkers concerns the discrimination between their prognostic or predictive value  in the response to immunotherapy. I don't know if the authors have examples of biomarkers that distinguish these 2 roles.

- It could sometimes be clarified whether the biomarkers mentioned are measured in baseline or during treatment and which ones remain predictive in multivariate analysis.

Author Response

We thank the reviewer # 2 for the time and effort put on revising our manuscript and for the positive feedback made on its format and content.

Some suggestions to improve the impact of this review

  • In the biomarkers of the tumor microenvironment, PD-L1 and TMB are well developed. TLS (Petitprez F Nature 2020) and resident memory CD8+T cells)(Karaki S et al J Immunother 2021) (Mami-Chouaib F et al J Immunother Cancer 2018)(Voabil P Nat Med 2021) could be mentioned.

Following the reviewer’s suggestion, in section “2. Circulating immune cells”

we have added:

“Interestingly, soft tissue sarcomas represent a heterogeneous group of cancers, with more than 50 histological subtypes. A gene expression study (n=608) of sarcomas has allowed a classification of 5 subtypes based on immunity from the composition of the tumour microenvironment. Subtype E (high immunity) characterized by the presence of tertiary lymphoid structures (TLS) containing T cells and follicular dendritic cells and particularly rich in B cells. Group E demonstrated improved survival and a high response rate to PD1 blockade with pembrolizumab in a phase 2 clinical trials. This work demonstrates the clinical potential of B-cell-rich TLS for decision-making and clinical treatments with ICs, which could have further potential applications in other cancers [31]. In addition, the baseline presence of tertiary lymphoid structures and their components correlated with the capacity of cancers to undergo intratumoral immune cell reactivation.

Finally, CD8+ T lymphocytes are the main anti-tumor effector cells. Most cancer immunotherapeutic strategy seeks to increase cytotoxic T lymphocytes (CTL) specific to tumoral cells. The tissue-resident memory T (TRM) cells, a recently identified subpopulation of memory CD8+ T cells, persists in peripheral tissues and does not recirculate. TRM cells subpopulation is considered an independent memory T-cell lineage with a particular transcription factors expression. TRM cells also express high levels of granzyme B, IFNγ and TNFα (see below), reinforcement their cytotoxic characteristics. In spontaneous tumor models and engrafted tumors, natural TRM cells or cancer-vaccine-induced TRM directly control tumor growth.  In addittion, TRM cells predominantly express immune checkpoint receptors such as Tim-3, PD-1 and CTLA-4. Blockade of PD-1 with anti-PD-1 antibodies on TRM cells obtained from human lung cancer promotes cytolytic activity toward autologous tumor cells. Therefore, TRM cells appear to represent relevant factor in tumor immune surveillance. TMR cells induction by immuno-therapy strategy may be crucial for the success of these treatments. TRM cells characteristics: 1-contact with tumor cells, 2-predominant expression of immune checkpoint receptors and 3- recognition of tumor cells, suggest that they may be involved in the success of ICIs treatment in various cancers [32].

J Immunother Cancer. 2021 Mar;9(3):e001948. doi: 10.1136/jitc-2020-001948.

CXCR6 deficiency impairs cancer vaccine efficacy and CD8 + resident memory T-cell recruitment in head and neck and lung tumors. Soumaya Karaki.           

This paper alludes to the stimulation of the immune system by the use of vaccines and not by ICIs, therefore, we have not considered it relevant to the topic of the review.      

  • At the blood level, there is a lot of recent work on Tcell -repertoire as a predictive biomarker of clinical response to immunotherapy. The authors cite the 2017 article by Akyüz. Other more recent articles could be reported (Anagnostou V CelL Rep Med 2021, Wu TD Nature 2020).

Following the reviewer’s suggestion, in section “2. Circulating immune cells”

we have added :

 “In a very recent study, Anagnostou et al. [29] performed comprehensive genomic and transcriptomic tumor analyses in conjunction with T cell repertoire analyses for 64 advanced melanoma patients treated with ICIs as part of the CheckMate-038 clinical trial (NCT01621490). Authors show that the tumor mutation burden is associated with improved treatment response, however, the mutation frequency in expressed genes is superior in predicting outcome. Increased T cell density in baseline tumors and dynamic changes in regression or expansion of the T cell repertoire during treatment differentiate responders from non-responders subjects. in addition, Transcriptome studio showed an increased abundance of B cell subsets in tumors from responders and patterns of molecular response associated to expressed mutation elimination or retention that reflect clinical outcome. Furthermore, they show that CD8+ T-cell exhaustion, identified here through TOX expression, could be reversible and lead to a therapeutic effect for ICIs.

In addition, Wu et al. [30] showed clear evidence of clonotypic expansion of effector-like T cells not only within the tumour but also in the surrounding healthy tissue. Subjects with gene signatures of such clonotypic expansion respond best to anti-PDL1 treatment. Remarkably, expanded clonotypes found in the tumour and in the surrounding healthy tissue can also be detected in peripheral blood, which suggests a suitable approach to patient identification.  These results and external datasets suggest that intratumoural T cells, mainly in responsive patients, are replenished with fresh, non-exhausted replacement cells from sites outside the tumour, suggesting continued activity of the cancer immunity cycle in these individuals, the acceleration of which may be associated with clinical response.”

3-Similarly, the authors could cite the work of I Melero on the value of IL-8 as a biomarker of response to immunotherapy (Schalper KA et al Nat Med 2020)

Following the reviewer’s suggestion, in section “7. Tumor necrosis factor, interferon gamma, and interleukins”, we have added:

“In a large-scale retrospective analysis (n=1344), Schalper et al. [53], demonstrated that elevated baseline serum IL-8 levels are associated with poor outcome in patients with advanced cancers (melanoma, NSCLC, renal cells carcinoma) with nivolumab and/or ipilimumab, everolimus or docetaxel therapy. This study showed the significance of evaluate serum IL-8 levels in identifying unfavorable tumor immunology and as an independent biological marker in subjects receiving ICIs.”

4-Line 125-129: the sentence "Other 125 studies have found that T-cell receptors (TCR) that allow for the recognition of a variety 126 of epitopes [26] are extremely similar in PDL1+ and CD8+ T cells and TILs, indicating that 127 PDL1 expression in peripheral T cells could be an indicator of the immune status of the 128 tumor tissue [25]" could be clarified.

Following the reviewer’s suggestion, in section “2. Circulating immune cells” we have added:

“Dronca et al [26], analysed peripehral blood levels of Bcl-2-like protein 11 (BIM) in patients with metastatic melanoma. They observed that patients with clinical benefit after four cycles of treatment with immunotherapy presented a higher frequency of Bim + / PD-1 + CD8 T lymphocytes at the beginning of the study compared to patients with disease progresion (mean of 60% vs. 49%, p = 0.04). In 9/9 responding patients, PD-1 + CD8 T-cell Bim levels decreased after the first three months of treatment and increased or did not change in the 5/5 non-responders (p = 0.003). Authors conclude that Bim levels in tumor-reactive PD-1+ CD8 T cells may select patients likely to benefit from anti-PD-1 therapy”.

5-It could be discussed that an important point in these biomarkers concerns the discrimination between their prognostic or predictive value  in the response to immunotherapy. I don't know if the authors have examples of biomarkers that distinguish these 2 roles.

We appreciate the reviewer's comment, and indeed we agree with his appreciation of the importance of specifying whether the biomarker is associated with the prediction of response to treatment, or with patient survival. Despite not being explicit, in the description made of each of the biomarkers analyzed in the different studies, it is mentioned whether it is associated with response (preactive biomarker) or survival (prognostic biomarker). Defining the type of biomarker is undoubtedly of interest, although we consider that focusing in the distingtion of these roles would modify the structure of our review.

6-It could sometimes be clarified whether the biomarkers mentioned are measured in baseline or during treatment and which ones remain predictive in multivariate analysis.

Again, we agree with the relevance of the point made by the reviewer. In our review we have tried to summarize the findings described in the different studies, pointing out the methodology used and giving importance to the time of measurement (baseline or sequential during treatment), and the presence of multivariate analysis that reinforce de value of the biomarker.
